# Biosynthetic constraints on amino acid synthesis at the base of the food chain may determine their use in higher-order consumer genomes

**Javier Gómez Ortega** [1], **David Raubenheimer** [2], **Sonika Tyagi** [1], **Christen K. Mirth** [1☯], **Matthew D. W. Piper** [1☯] *

1 School of Biological Sciences, Monash University, Clayton, Victoria, Australia, 2 The University of Sydney, Charles Perkins Centre and School of Life and Environmental Sciences, Sydney, Australia

☯ These authors contributed equally to this work.
* matthew.piper@monash.edu

**Data Availability Statement:** All data files are available from Figshare (https://doi.org/10.26180/15048150.v2).

## Abstract

Dietary nutrient composition is essential for shaping important fitness traits and behaviours. Many organisms are protein limited, and for *Drosophila melanogaster* this limitation manifests at the level of the single most limiting essential Amino Acid (AA) in the diet. The identity of this AA and its effects on female fecundity is readily predictable by a procedure called exome matching in which the sum of AAs encoded by a consumer's exome is used to predict the relative proportion of AAs required in its diet. However, the exome matching calculation does not weight AA contributions to the overall profile by protein size or expression. Here, we update the exome matching calculation to include these weightings. Surprisingly, although nearly half of the transcriptome is differentially expressed when comparing male and female flies, we found that creating transcriptome-weighted exome matched diets for each sex did not enhance their fecundity over that supported by exome matching alone. These data indicate that while organisms may require different amounts of dietary protein across conditions, the relative proportion of the constituent AAs remains constant. Interestingly, we also found that exome matched AA profiles are generally conserved across taxa and that the composition of these profiles might be explained by energetic and elemental limitations on microbial AA synthesis. Thus, it appears that ecological constraints amongst autotrophs shape the relative proportion of AAs that are available across trophic levels and that this constrains biomass composition.

## Author summary

The amount and type of food that organisms consume shapes their fitness. Many species, including the fruitfly *Drosophila melanogaster*, suffer protein-limitation, which means they must evolve strategies to make the most of the protein they consume. We previously

**Funding:** This work was funded in part by the ARC (FT150100237), the NHMRC (1182330) to M.D.W. P., as well as the ARC (FT170100259) to C.K.M. (ARC is the Australian Research Council - www. arc.gov.au; NHMRC is the National Health and Medical Research Council - www.nhmrc.gov.au). The funders had no role in study design, data collection and analysis, decision to publish, or preparation of the manuscript.

**Competing interests:** The authors have declared that no competing interests exist.

discovered that this protein limitation manifests at the level of individual amino acids for egg production in fruitflies.

Here, we attempt to improve dietary amino acid proportions (protein quality) for male and female reproduction in the fruitfly. We do this by tailoring the fly's diet to contain amino acids in the proportions found in all the expressed proteins of either male or female *Drosophila*. In doing so, we discover that, despite functional differences between the sexes, their pattern of genome-encoded amino acid utilisation is remarkably conserved. In fact, this amino acid profile is also conserved in other species' genomes from bacteria to humans. We hypothesise that this conservation represents an evolutionary strategy for organisms to make the most of limited amounts of dietary protein.

## Introduction

Nutrition is one of the most important environmental determinants of evolutionary fitness; it supplies organisms with energy and the building blocks they require for growth, reproduction, and somatic maintenance [1]. However, the natural availability of food and its nutritional qualities vary and inevitably differ from the consumer's needs [1–3]. As such, evolutionary fitness is constrained by the divergence between nutrient demand and their availability. Because of this, optimising nutrition to enhance growth, reproduction, and health is of major interest from both a fundamental biology and a commercial perspective.

Among the main components in the diet, protein is the limiting nutrient for the growth and reproduction of many organisms. It is, therefore, a principal constraint on evolutionary fitness [1,4–6]. For example, the abundance of protein-rich food has been shown to increase population size or stimulate body growth of birds such as the galah (*Eolophus roseicapilus*) or the goldfinch (*Carduelis carduelis*) and mammals like the house mouse (*Mus musculus*) or several species of squirrels (*Sciurus and Tamiasciurus spp.*) [7–13]. In the fruit fly *Drosophila melanogaster*, we have found that female reproduction is reduced by decreasing overall dietary protein concentration [14–16]. We also found that this protein limitation is determined by the concentration of the single most limiting essential Amino Acid (AA) in the diet, which can be identified by comparing the proportion of AAs that is available in food against the proportion of AAs encoded by the fly's exome–a procedure we called exome matching [14–16]. Evidence from our work, and that of others, indicates that exome matching may have broader application as protein limitation also occurs at the level of single AAs in other species [14,17–20].

To predict limiting AAs, our exome matching protocol involves two steps. First, we calculate each AA's relative abundance in every protein of an organism's *in silico* translated exome. Second, we find the average proportional representation for each AA across all proteins encoded by the exome. This genome-wide averaged AA proportion can then be compared to the AA proportion in the food to identify the essential AA that is most underrepresented in the diet and thus predicted to be limiting. We demonstrated that supplementing the diets of flies and mice with the limiting AA that was identified in this way can improve growth and reproduction and modify feeding behaviour [14,21,22]. Thus, for every organism whose genome has been sequenced, exome matching can theoretically be used as a tool to guide precision nutrition for better health.

Although we showed exome matching to be biologically effective, its current implementation does not incorporate weightings for the substantial differences we know to exist in genes' sizes and their degree of expression [23]. Many studies have documented considerable differences in gene expression profiles when comparing transcriptomes between sexes, across life-

history stages, or in response to biotic and abiotic stimuli [24–28]. For instance, in *Drosophila*, more than 8,000 genes, representing at least 50% of the genome, have been reported to be differentially expressed when comparing adult males with fertilised females–an observation that is unsurprising given the much heavier anabolic burden of reproduction for females than for males [25]. Thus, we predicted that we could improve the precision of exome matching by incorporating weightings for gene expression changes and in doing so, would establish a new way of tailoring diets to match an organism's individual AA demands for life-stage and health status. Here, we set out to test this prediction and, in doing so, uncover that there is a surprisingly small variation in the way transcriptome weightings modify predicted AA usage. These data may reflect fundamental energetic and nutrient constraints on body composition across taxa.

## Results

### Calculating transcriptome-weighted, exome-matched dietary aa proportions

Our previous research demonstrated that female flies fed food containing exome-matched AA proportions (FLYAA) laid more eggs than flies on food with equivalent amounts of protein comprised of mismatched AA proportions [14]. Although FLYAA demonstrably improved egg-laying, we hypothesised it could be further improved by weighting each gene's contribution to the overall average by its length and expression level. We reasoned that although there is not a 1:1 association between transcription and translation, the transcriptome would be a good approximation for the expression weightings for two reasons. First, transcriptomics readily yields a more complete set of gene expression values than proteomics [29,30]. And, second, if the availability of dietary AAs constrains organismal protein expression, whole genome proteomics would simply reflect the constraints of diet quality. In contrast gene expression values may indicate protein expression levels that could be achieved if dietary AA availability was not a constraint—i.e. better matched to requirements.

We downloaded transcriptome profiles of whole male and whole female flies from FlyAtlas 2 and modENCODE, and from these profiles we averaged the levels of gene expression for each sex [31,32]. To make our new sex-specific, transcriptome-matched profiles, we first counted the number of each AA that is encoded by each protein isoform in the fly genome. We then weighted these AA counts by the average isoform relative transcript abundance (FPKM value; see Materials and Methods) found for male or female flies. For each AA, we then summed the weighted AA counts across all genes and used these values to compute each AA's proportional representation across all expressed genes. Although we observed some differences between the transcriptome profiles obtained from FlyAtlas 2 and modENCODE, they produced concordant changes in the proportion of each AA when compared with FLYAA. These newly designed AA ratios for the sexes were labelled MALEAA and FEMALEAA, and these became the basis for new dietary AA profiles (Fig 1).

### Transcriptome-weighted exome matching the dietary aa profile does not improve male fertility over that on an exome matched diet

To test the effects of dietary AAs on male fertility, we used an assay in which males were challenged to inseminate females at maximum capacity, as this should deplete the males of sperm and/or seminal fluid and thus require them to be synthesising more from the dietary AAs they have available. To do this, we supplied singly housed males with ten new virgin females per day for seven days and counted how many of these females subsequently produced viable

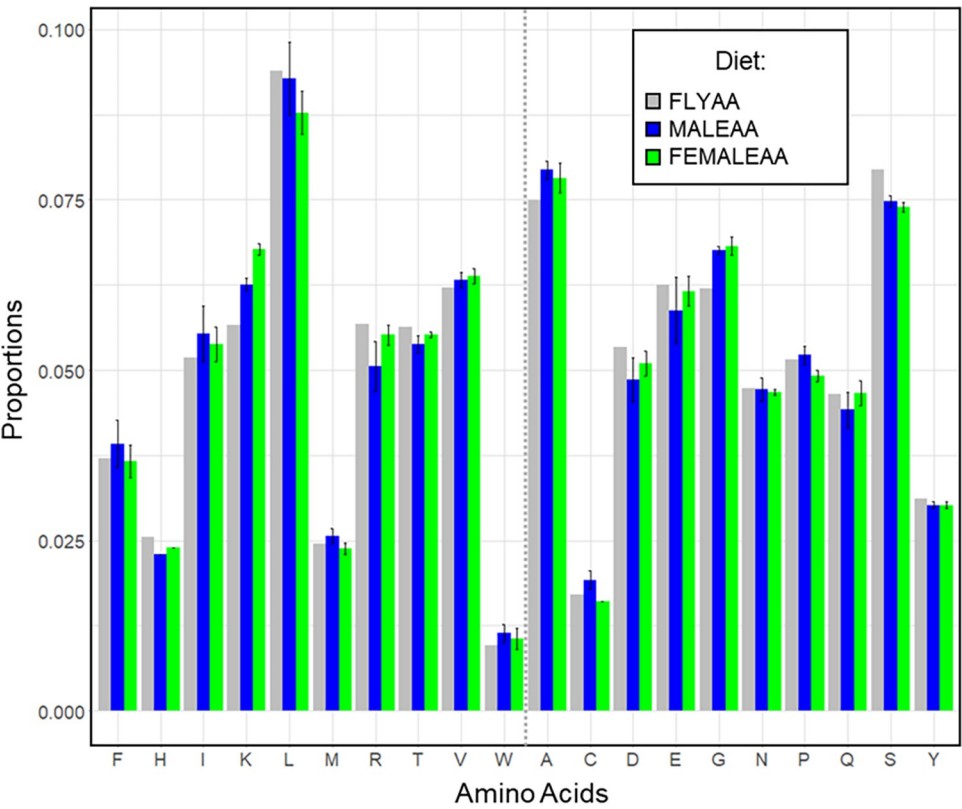

**Fig 1. Comparison of the exome matched and transcriptome weighted AA ratios.** Proportion of each AA in the exome matched [14] and transcriptome-weighted exome matched diets (MALEAA and FEMALEAA). In these diets, the average proportions of AAs have been generated after weighting each protein's contribution by size and its average gene expression in male and female transcriptomes, respectively. The dotted grey line separates the essential (left) from the non-essential (right) AAs. IUPAC single-letter AA codes are shown. Error bars display the standard deviation from weighting the AA ratios by each of the replicate transcriptome profiles from FlyAtlas 2 and modENCODE that were available for males or females.

offspring. We found that while on the first day, males could inseminate 8 to 10 of these virgins, during the course of the assay, the number of females that each male could inseminate dropped to at least half of the number found for day one (Fig 2A) indicating that the males were indeed operating at maximum capacity in this assay.

To test if male fecundity changed in response to altered dietary protein levels, we performed the above assay on males that were maintained on food in which the AA proportion was fixed (FLYAA), but the total concentration was diluted from 10.7g/l (positive control; the level in our standard "rich" diet) to 2.1 g/l, 1.1 g/l and 0 g/l. Male fertility was significantly lower than the positive control when the flies were maintained on 0 g/l and 1.1 g/l AAs (P<0.001) (Fig 2A). Surprisingly, when the protein concentration was 2.1 g/l, male fertility increased to the level of flies maintained on the positive control condition (Fig 2A and 2B). Thus, maximum male fertility in our assay responded to dietary AA levels and only relatively small amounts were required to support maximal fertility. The response of male fecundity to dietary AA concentration could be modelled by a sigmoidal dose-response curve with an inflexion point somewhere between 1.1–2.1 g/l AA (Fig 2B).

If MALEAA represents the ideal proportion of AAs for male fecundity, we predict that males fed FLYAA would be tryptophan (trp, W) limited, and that for a fixed sum of AAs changing the proportion to MALEAA would yield a 20% increase in AA availability for

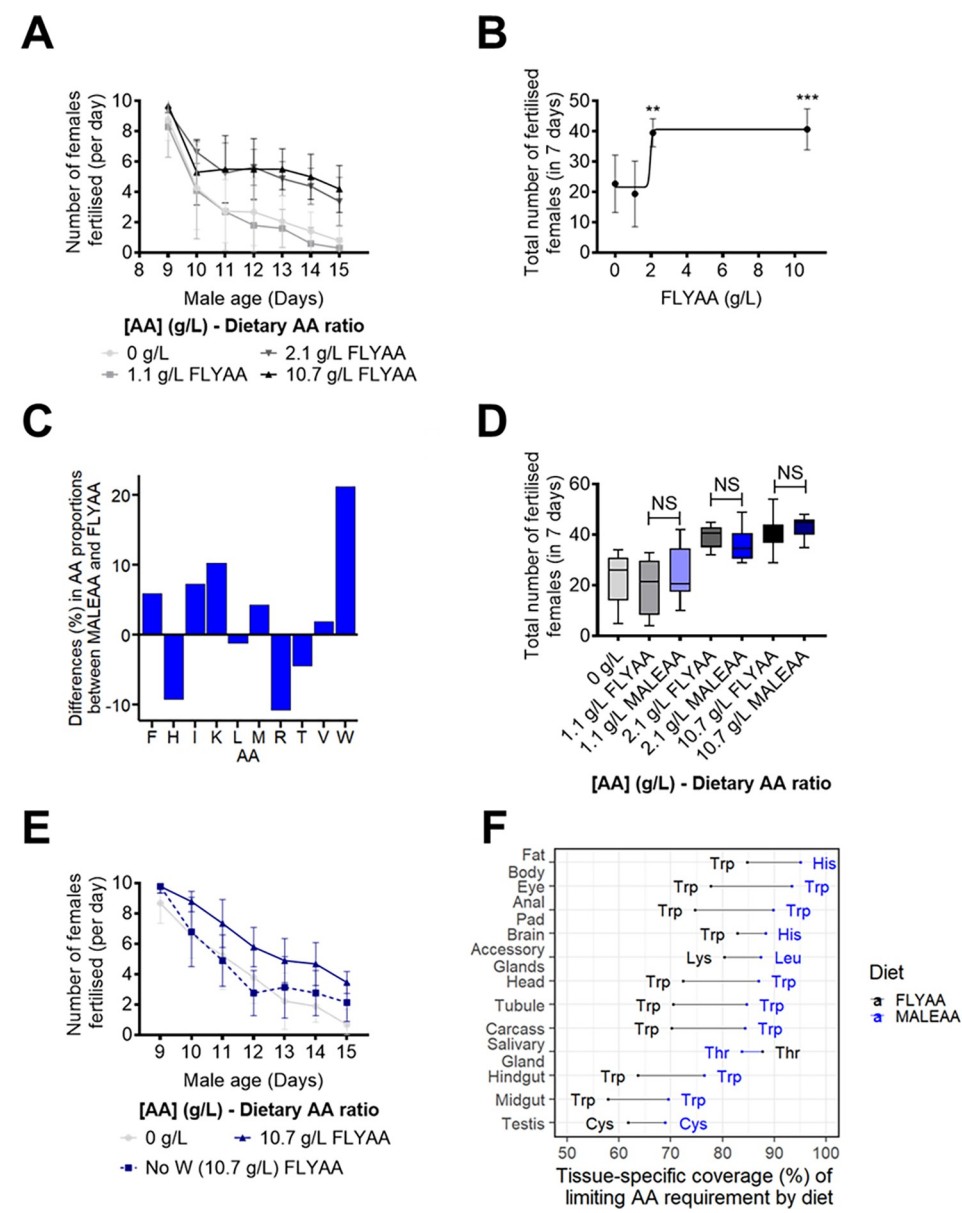

**Fig 2. Male fecundity is modified by dietary AA concentration but not ratio.** (A). The number of virgin female flies successfully fertilised by individual males during our seven-day assay. Male fecundity was reduced by decreasing dietary AA concentrations. AAs were provided in the exome matched proportion, FLYAA). Error bars represent the standard deviation. (B). The change in the cumulative number of females fertilised by males in response to dietary AA change could be modelled by a sigmoidal dose-response curve ($R^2$ = 0.537; least-squares fit). (*** = P<0.001, in comparison with 0 g/L). Error bars represent the standard deviation. (C). Predicted difference in each essential AA when comparing the male transcriptome matched proportions (MALEAA), and the exome matched proportions (FLYAA). A positive difference indicates that the AA is more abundant in MALEAA than in the FLYAA. MALEAA should cover any essential AA deficiency of FLYAA, and thus, the relative increase in the concentration of the most limiting essential AA (Tryptophan, W, 20%) equals the potential increase in fecundity that could be achieved for flies fed with MALEAA. (D). Males fed with a diet containing a transcriptome (MALEAA) matched AA proportion did not differ from those fed the exome matched (FLYAA) diets for any concentration of AAs tested. Error bars represent the standard deviation. (E). The effect of tryptophan dropout from the diet on the daily capacity of males to fertilise females during a seven-day period. The removal of tryptophan from the diet caused a fast decay in fertilisation that matched that caused by the removal of all AAs. AAs were provided in the male transcriptome matched proportion, MALEAA. Error bars represent the standard deviation. (F). Coverage of the predicted dietary AA requirements by MALEAA and FLYAA when compared to the transcriptome weighted exome match proportion from each tissue in male flies. For each tissue, the x-axis displays the degree to which the limiting AA demand is met by the diets MALEAA (blue) and FLYAA (black). The closer to 100%, the better the diet covers the predicted tissue demand for

AAs. For all tissues, except the salivary gland, MALEAA is predicted to be a better match for requirements than FLYAA. The predicted limiting AA for each tissue on each diet is indicated by the three letter AA codes.

reproduction (Fig 2C). However, when we compared the fecundity response of male flies kept on MALEAA and FLYAA at each of the dietary AA concentrations, we saw that AA ratio did not alter the number of females that were successfully inseminated, even under conditions where male fertility was clearly AA limited (1.1 g/l: Fig 2D). This lack of effect was not due to an insufficient sampling since a power analysis revealed that a 20% difference in fecundity would have been observable at 2.1 g/l.

A possible reason why MALEAA did not improve fecundity is that males might contain sufficient stores of tryptophan in body proteins that they can retrieve and use to overcome the limitation we predicted. If this were the case, our prediction of a 20% improvement in fecundity would be an overestimate. To assess this, we made another diet in which the AA profile resembled FLYAA, but tryptophan only was omitted from the diet altogether. We evaluated the effect of this diet and found that it caused a significant reduction in fecundity compared to the positive control diet (10.7g/l). It was also equally as detrimental for fecundity as a diet without AAs, both in terms of the rate at which fertility fell, and the total number of females successfully fertilised during the assay (Fig 2E). Thus, dietary tryptophan is required to sustain male fecundity in this assay, and its requirement does not appear to be lessened due to the recovery of tryptophan stored in body tissue.

Another reason why MALEAA may not have improved fecundity over FLYAA is that the actual set of proteins required for male fecundity are only a subset of those included in our calculation and that MALEAA is actually a worse AA balance for the organs responsible for making the proteins required for fecundity. To investigate this possibility, we calculated transcriptome-weighted AA profiles for each tissue type in male flies using RNAseq data from FlyAtlas [31]. We then compared these tissue profiles to both the unweighted (FLYAA) and transcriptome weighted (MALEAA) dietary AA profiles and predicted each tissue's limiting AA and the degree to which it is limiting. The data are expressed as a relative match where 0 indicates the complete absence of an essential AA and 100 represents that all dietary AAs are perfectly matched to the tissue-specific profile (Fig 2F). The data show that MALEAA is predicted to be a better match than FLYAA for the expression of the genes in each tissue, except for those in the salivary glands in which FLYAA is predicted to be a slightly better match than MALEAA. Thus, unless AA supply to the salivary glands limits whole organism fecundity we still predict that MALEAA would be an improved AA profile over FLYAA for male reproduction if transcriptome weighting the exome provided a superior prediction of dietary requirements. However, our tissue-specific analysis does reveal that MALEAA is predicted to confer a smaller improvement over FLYAA if only the profile of the testis (9.6% increase) or accessory glands (10% increase) matter for our assay of male fecundity. It is possible that this small degree of enhancement in male fecundity was beyond the sensitivity of our assay to be detected.

### Transcriptome-weighted exome matching the dietary aa profile does not improve female fecundity over that on an exome matched diet

Our previous data indicate that female egg-laying is a reliable indicator of dietary AA composition, and typically has lower variability and greater sensitivity than the male fecundity assay we performed [14]. We thus tested if FEMALEAA had a higher nutritional value than FLYAA to sustain female fecundity. Two-day-old mated females were placed on chemically defined diets, and the number of eggs they laid over the course of eight days was counted. In line with previous results [14], female egg production responded in a linear manner to increasing AA

concentrations until at least 10.7 g/L (Fig 3A). This is consistent with dietary AAs quantitatively limiting female egg-laying, which we have previously shown to be due to the most limiting essential AA [14].

We predicted that if the transcriptome-weighted diet (FEMALEAA) represented the actual AA requirements for females for egg-laying, lysine (K, Lys) would limit egg production for females feeding on the non-weighted AA ratio (FLYAA) (Fig 3B). By comparing the AA profile of FEMALEAA to that of FLYAA, we also predicted that egg production could be up to 20% higher when incorporating transcriptome weightings into the exome match profile (Fig 3B). However, FEMALEAA did not improve female fecundity output in comparison to FLYAA at either concentration of dietary AAs tested (Fig 3C). This included a concentration at which AAs clearly limited egg production (5.4 g/L) and so should be the most sensitive test of the change in availability of the most limiting AA. This lack of effect was not due to insufficient sampling since a power analysis revealed that a 20% difference in fecundity would have been observable at 5.4 g/l and 10.7 g/l AA.

We tested if the reason why female flies produced no more eggs on FEMALEAA than when on FLYAA is that they could supplement limiting dietary lysine by retrieving it from body protein reservoirs. If dietary lysine limits egg-laying, and the flies do not supplement it from body reserves, the flies should exhibit reduced egg production in proportion to the dilution of lysine in the diet. This is exactly what we observed when we reduced lysine only in an otherwise constant nutritional background containing 10.7 g/L FLYAA (Fig 3D). This demonstrates that lysine is both essential and limiting in FLYAA for female fecundity and indicates that lysine limitation is not lessened by flies retrieving it from stored body protein.

We also tested if FEMALEAA is not superior to FLYAA to support egg production because the flies use only a subset of the transcriptome to produce eggs. To do this, we assessed the match between each tissue-specific AA profile and FEMALEAA or FLYAA. Similar to the comparison we made for males, FEMALEAA was a better match than FLYAA to the transcriptome weighted AA proportions of every female tissue except for the salivary glands (Fig 3E). Furthermore, if we consider only the tissues most relevant to reproduction, the ovaries and fat body, FEMALEAA represents a better match, and to a similar extent as whole-body samples, to their transcriptome-weighted exome profiles than FLYAA (16% and 15% for ovaries and fat body respectively). These data support our prediction that FEMALEAA should improve dietary AA availability for egg production over that found in FLYAA.

## Metabolic costs of amino acid biosynthesis may constrain their relative abundance in the biomass composition of higher order consumers

We have found that weighting the dietary exome matched AA ratio (FLYAA) by the average gene expression of male (MALEAA) or female (FEMALEAA) flies neither substantially modified the predicted porportions of AA used by the flies to optimise performance (Fig 1) nor modified their actual requirements for fecundity. This was surprising because the weightings incorporated changes in ~50% of the expressed transcriptome. To assess the generalisability of this observation, we also generated the AA proportions for both transcriptome-weighted and non weighted exome data for flies that were challenged with infection—a major physiological stressor that is known to elicit a substantial transcriptional response [33]. Again, we found that there was no notable change in AA usage over the time course of infection and the flies' predicted AA usage was very similar to controls (S1 Fig).

These data indicate that despite large changes in gene expression, there are constraints on the degree to which genome-wide expression changes can modify the dietary AA requirements of our flies. One of the ways this could happen is if the male and female transcriptome-

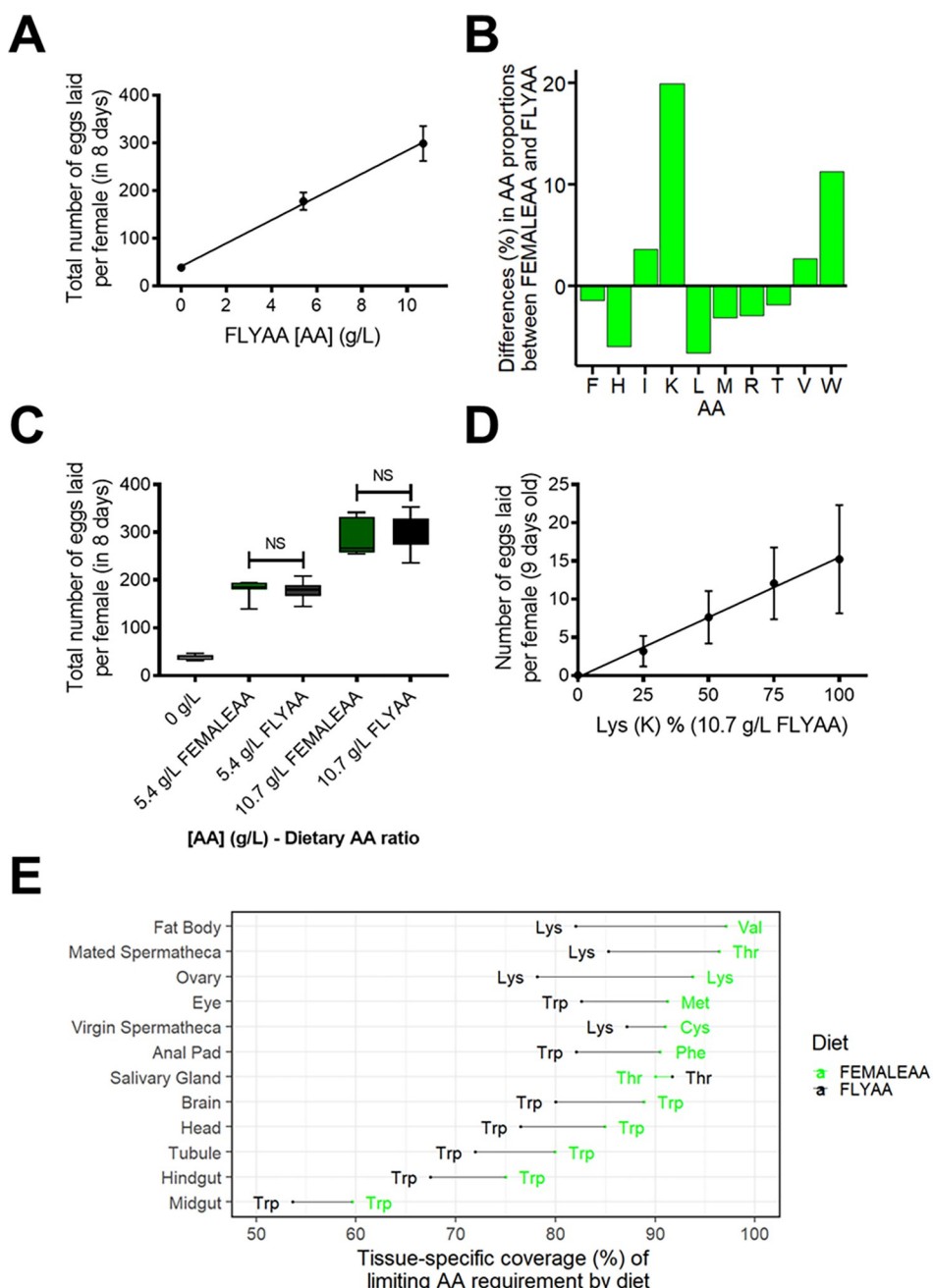

**Fig 3. Female fecundity responses to changes in dietary AA ratio and concentration.** (A). Total female fecundity showed a linear response to increasing dietary AA concentrations (FLYAA). $R^2$ = 0.999. (B). Relative change in the molar concentration of each essential AA between the female transcriptome matched diet (FEMALEAA) and the exome matched diet (FLYAA). A positive difference indicates that the AA is more abundant in FEMALEAA than in FLYAA. The relative increase in the concentration of the most limiting essential AA (Lysine, K, 20%) equals the potential increase in fecundity that could be achieved for flies fed with FEMALEAA. (C). At each concentration of total AA, there was no difference in egg-laying output of females fed with transcriptome (FEMALEAA) or exome (FLYAA) matched diets. (D). Lysine dilution limits egg production in a linear manner. Percentages of Lysine concentration are relative to the standard Lysine concentration on FLYAA with a total AA concentration of 10.7 g/L. R2 = 0.995. (E). Coverage of the predicted dietary AA requirements of each tissue in female flies by FEMALEAA and FLYAA. Tissue dietary AA requirements are predicted from the tissue-specific transcriptomes. For each tissue, the x-axis displays the percent of the limiting AA demand covered by the diets FEMALEAA (green) and FLYAA (black). The closer to 100%, the better the diet meets the theoretical tissue AA demand. For all tissues, except the salivary gland, FEMALEAA is predicted to be a better match for requirements than FLYAA. The predicted limiting AA for each tissue on each diet is indicated by the three-letter AA codes.

weighted AA profiles converge on a similar AA usage as the non-weighted (FLYAA) profile. To assess the degree of similarity of our three AA profiles (FEMALEAA, MALEAA and FLYAA), we compared them in the context of a null distribution of expression profiles. To generate this null distribution, we permuted the gene labels on the male and female transcriptome-wide expression data 20,000 times, thus generating a set of divergent but biologically realistic transcriptome-wide expression profiles. For each of these profiles, we then calculated the transcriptome-wide AA usage, and then calculated the Euclidian distance between each of the permuted AA profiles and the median permuted profile. This revealed that the AA proportions of FLYAA, FEMALEAA and MALEAA were more distant from the median transcriptome-weighted AA profile than most of the permutations (>99% for FEMALEAA, >92%, for MALEAA, and >97% for FLYAA in the context of FEMALEAA and >74% for FLYAA in the context of MALEAA), indicating that, compared with the null distribution, they all represented extreme examples of AA usage (Fig 4A and 4B). Because Euclidian distance only measures the degree of divergence between profiles and not the direction of individual AA change (i.e. whether an AA has higher or lower representation), we plotted the proportional representation of each AA for each AA profile (Fig 4C and 4D). When we did this, we found that for the AAs where MALEAA and FEMALEAA lay beyond the limits of the box plots, thus differing from the majority of permuted values, FLYAA tended to differ from the permuted values in the same direction. Thus, the actual transcriptome weighted AA profiles that we found for males and females used AA in a manner more similar to the non-weighted AA profile (FLYAA) than the majority of the permuted profiles. This could indicate a constraint on the way that the transcriptome can modify the consumer's AA requirements.

To assess how broadly these constraints might be found, we generated exome matched AA proportions for a bacterium, a yeast, flies, mice, corn and humans (S2 Fig). Despite the evolutionary divergence of these organisms, their profile of AA usage encoded by their genomes appears to be similar, perhaps indicating that constraints on AA usage is broadly conserved.

## Discussion

Organisms live in a changeable environment in which the amount and quality of their food varies [7]. These variations can limit fitness by restricting the supply of energy and building blocks that are needed for growth, reproduction, and to sustain health [1,7,34]. Consumers have therefore evolved numerous behavioural, symbiotic, biochemical, and physical adaptations to buffer against nutrient supply variations [1,5,7,21,35–39]. In a previous study, we found that the AA proportions encoded by the *Drosophila melanogaster* exome provided a good definition of dietary protein quality for reproduction–a procedure we called exome matching [14]. In this study, we have made the somewhat surprising finding that despite highly divergent levels of gene expression between the sexes in *Drosophila*, adjusting their exome matched diets to include weightings for male and female transcriptome profiles: 1) does not markedly change their predicted AA requirements from that determined by exome matching alone (i.e. without gene expression weightings), and; 2) does not modify female and male reproduction over that found when flies feed on exome matched dietary AA proportions. These effects indicate that the expressed fly proteome utilises AA on average in a relatively constant manner, perhaps revealing a strategy to utilise limiting amounts of dietary protein with greater efficiency. This strategy would be effective when the AA proportions found in the flies' diet remains relatively constant and similar to that found in fly biomass.

Heterotrophs ultimately rely on the availability of organic molecules that are supplied to them by autotrophs. If AA proportions are relatively constant across organisms, their profile may reflect constraints on AA production at the level of autotrophs. Interestingly, Lightfield

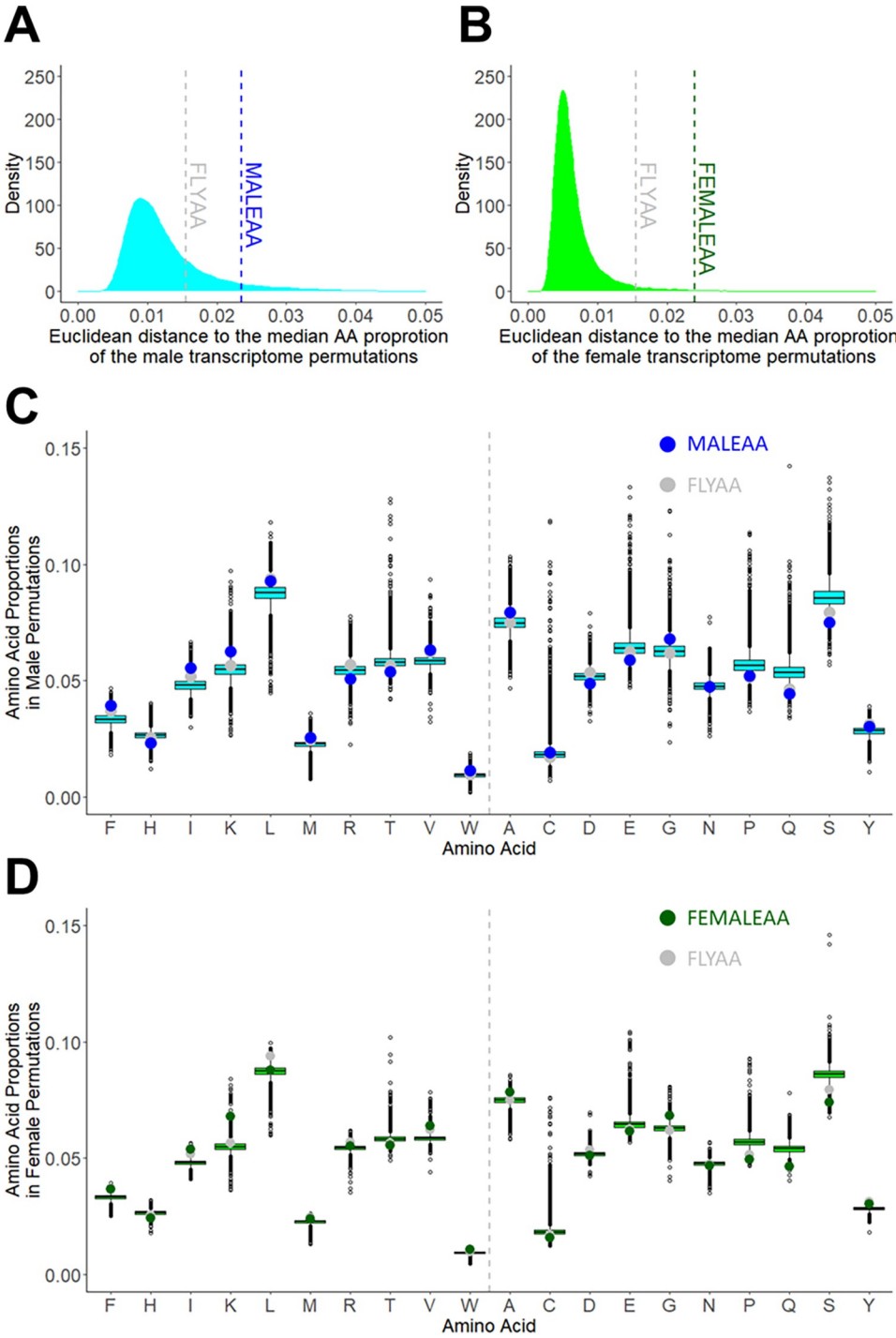

**Fig 4. Using permuted transcriptome-weighted AA profiles to compare the way in which the sex-specific transcriptome weighted profiles differ from exome matching.** (A, B). We permuted the gene labels in our male (A) and female (B) transcriptome data 20,000 times and calculated a new transcriptome-weighted AA profile for each permutation. We then calculated the Euclidean distances from the median AA proportion of the permuted profiles to each permutation from the male (cyan) (A) and female (light green) (B) transcriptome data as well as the distance to MALEAA (dashed blue line) (A), FEMALEAA (dashed green line) (B) and FLYAA (dashed grey line) (A,B). (C, D) The relative proportion of each AA for all permuted profiles as well as the proportions found for MALEAA (blue) (C), FEMALEAA (green) (D), and FLYAA (grey) (C, D). The vertical grey dashed line separates essential (left) from non-essential (right) AAs. The median AA values are represented by a horizontal line dividing the boxes. The boxes represent the interquartile range (50% of the permuted values) and points shown are values at least 1.5-times the upper and lower interquartile values.

et al. [40] found that despite large changes in genomic GC composition, a broad range of bacterial taxa still contain a relatively constant profile of genome encoded AA usage. Here, we have found that this AA profile also appears to be conserved across diverse organisms from bacteria to humans. These data support the idea that the relatively constant proportion of AA in biomass result from production-level constraints that are reflected up the trophic chain. But what determines the particular AA profile that we observe?

Using data from more than 100 organisms across three domains of life, Krick et al. [41] described a significant negative association between the relative abundance of an AA encoded by a genome and its metabolic costs of production and use. The authors propose that this is the result of an energetic balancing act, in which the overall costs of AA synthesis are minimised, while appropriate levels of sequence diversity, which ensures proteome stability, is maintained [41–43]. While their regression represents an extraordinarily good fit, some unexplained variability appears for the abundance of a subset of AA, which indicates additional limitations beyond energy supply. In particular, the sulfur-conatining AA, which have been shown to be limiting in other studies, vary in microbial proteomes in a manner that reflects the environmental availability of sulfur [44,45]. In other work, we have also found evidence that flies experience a limitation for the sulfur containing AA methionine for egg production when feeding on their natural food source, yeast [46]. Moreover, we have found that female flies can specifically buffer against dietary methionine shortages by continuously producing viable eggs when methionine levels drop [47,48]. Together, these data support a model in which energetic, elemental and other nutrient supply constraints on autotrophs shape the AA composition of biomass in heterotrophs.

If limitations on autotrophs shape the AA proportions of higher order consumers, we expect that when moving up trophic levels, heterotrophs may contain more similar AA profiles to their food. This is because they generally only synthesise 10 or 11 AAs *de novo*, while autotrophs can synthesise all 20, which means higher order consumers cannot metabolically buffer against dietary AA variations as readily as producers. If correct, we anticipate that the largest discrepancies between the supply and demand of specific AAs would exist for organisms feeding at the interface of producers and consumers. This means that supplementing diets with essential AA should have the greatest effects to augment consumer fitness for fungivores and herbivores. Where these experiments exist, the data appear to support this prediction [7,8,11,13,18,46,49–51].

Understanding the way organisms interact with their nutritional environments has been the topic of an enormous body of research. Historically, the models employed to study ecosystem dynamics have employed a single currency (energy), but work over the last 30 years has described and modelled the role of other nutritional components in ecological constraints and organismal fitness [1,52,53]. Ecological Stoichiometry and Nutritional Geometry are two such models that provide complementary approaches to describing these systems [34,54,55]. Ecological Stoichiometry explains individual and ecosystem-level phenomena as shaped by the availability of energy and the chemical elements (e.g. carbon, nitrogen and phosphorous) that make up biomass [53]. By contrast, Nutritional Geometry models the same phenomena from the point of view of the biochemicals (e.g. carbohydrates and proteins) that modify feeding behaviour and evolutionary fitness [56]. Our data provide evidence for an association between the energetic and elemental constraints on autotrophs in shaping the proportions of the biochemicals (AAs) whose limitation subsequently shapes the composition of heterotroph biomass. These data resonate with the interesting argument that the elemental composition of life is closely correlated with element availability which suggests that through evolution, life has pieced together what was available rather than selectively composed an optimal mix of building blocks [57]. They also underscore the importance of considering the most relevant

constraints of the trophic level under study and highlight the need for emerging models that seek to combine the two approaches [54,55,58].

Part of the motivation of our study was to generate a new way of tailoring dietary AA profiles to meet the changing demands of organisms as they progress through development and experience altered health. Here we find no theoretical or experimental evidence to undertake such measures since the organism's AA requirements for its expressed genome tend to converge on that encoded by the exome. While we have used an example where broad scale transcriptome differences do not support evidence for changes in a phenotype that we already knew to be AA sensitive (female egg production), we do acknowledge that our experimental data do not rule out the potential benefits of transcriptome weighted exome matching in other contexts. Whatever the case, our study demonstrates that exome matching can serve as a simple guide to establish the dietary AA requirements of animals and so spare some of the expensive, time-consuming, and physically invasive efforts that are employed to determine the optimal dietary AA requirements of animals in agriculture [59,60] and for humans for health [61].

## Materials and methods

### Diets

**Sugar/yeast (sy) food.** Our sugar-yeast (SY) food contained sugar (Bundaberg, M180919) (50 g/L), brewer's yeast (MP Biomedicals, 903312) (100 g/L), agar (Gelita, A-181017) (10 g/L), propionic acid (Merck, 8.00605.0500) (3 mL/L) and nipagin (Sigma-Aldrich, W271004-5KG-K) (12 g/L), prepared as in [16]. Refer to Table 1 for more detailed product information.

**Holidic diets.** Chemically defined (holidic) diets used the recipe and were prepared as in [14] (Table 2). In all cases, the concentrations of all nutrients, except AAs, were held constant at the levels published in [14] (Tables 2, 3 and 4). The AA ratio was either matched to the fly exome, FLYAA [14], or the transcriptome-weighted dietary AA ratios (Table 5). Refer to Tables 3 and 4 for more detailed product information.

### Fly strain and conditions

All experiments were carried out using our outbred wild-type strain of *Drosophila melanogaster* called Dahomey [62]. Stock and experimental flies were kept under controlled conditions of 25°C with at least 65% relative humidity for a 12:12 h photoperiod. Stocks were maintained on SY food.

### Male fecundity assay

All flies were reared under standard density conditions as detailed in Bass et al., 2007 [16]. Males and virgin female flies were collected from 0 to 5 h after emerging and kept separately in fresh

**Table 1. Product amounts in 1000mL of Laboratory's Sugar-Yeast (SY) food.**

| Product | Amount | Unit | Company | Code |
|---|---|---|---|---|
| Agar | 1.00E+01 | g | Gelita | A-181017 |
| Graded Sugar | 5.00E+01 | g | Bundaberg | M180919 |
| Brewer's Yeast | 1.00E+02 | g | MP Biomedicals | 903312 |
| Nipagin | 1.50E+00 | g | Sigma-Aldrich | W271004-5KG-K |
| Propionic Acid | 3.00E+00 | ml | Merck | 8.00605.0500 |
| Absolute Ethanol* | 1.50E+01 | mL | Thermo-Fisher | AJA214-2.5LPL |
| *Solvent for nipagin | | | | |

**Table 2. Cooking instructions for 1000mL of Holidic Food.**

| Total mass of amino acids: 21.4 g | | | | |
|---|---|---|---|---|
| Diet: | FLYAA | MALEAA | FEMALEAA | |
| 1) Add 500 mL water | | | | |
| 2) Add the following amounts of the specified compounds and solutions | | | | |
| Agar | 20 | 20 | 20 | g |
| Ile | 1.12 | 2.24 | 2.16 | g |
| Leu | 2.03 | 3.76 | 3.55 | g |
| Tyr | 0.93 | 1.68 | 1.69 | g |
| Sucrose | 17.12 | 17.12 | 17.12 | g |
| Cholesterol Solution | 15 | 15 | 15 | ml |
| Buffer Solution | 100 | 100 | 100 | ml |
| CaCl2 (1000x Solution) | 1 | 1 | 1 | ml |
| MgSO4 (1000x Solution) | 1 | 1 | 1 | ml |
| CuSO4 (1000x Solution) | 1 | 1 | 1 | ml |
| FeSO4 (1000x Solution) | 1 | 1 | 1 | ml |
| MnCl2 (1000x Solution) | 1 | 1 | 1 | ml |
| ZnSO4 (1000x Solution) | 1 | 1 | 1 | ml |
| 3) Add water until reaching a total food volume of: | | | | |
| | 806 mL | 803 mL | 804 mL | |
| 4) Autoclave | | | | |
| 5) Add the following amounts of the specified solutions | | | | |
| Nucl/Lipid Solution | 8 | 8 | 8 | ml |
| EAA Stock Solution | 60.51 | 60.51 | 60.51 | ml |
| NEAA Stock Solution | 60.51 | 60.51 | 60.51 | ml |
| Glu (100mg/ml) | 15.19 | 15.48 | 16.24283 | ml |
| Cys (50mg/ml) | 6.83 | 9.72 | 8.022079 | ml |
| Vitamin Solution | 21 | 21 | 21 | ml |
| Folic Acid Solution | 1 | 1 | 1 | ml |
| Propionic Acid | 6 | 6 | 6 | ml |
| Nipagin Solution | 15 | 15 | 15 | ml |

standard SY food vials. When flies were two days old, male flies were transferred to chemically defined diets with different AA ratios (FLYAA or MALEAA) and different total AA concentrations (0 g/L, 1.1 g/L, 2.1 g/L or 10.7 g/L). Following a seven-day adaptation period, individually housed males were placed with ten new age-matched (+/- 1 day) virgin females. After being co-housed for 24h, the females were removed and replaced with another ten virgin females. This procedure was repeated every day for seven days. All females that were removed from the vial containing the male were singly housed in a new vial containing SY food. After ten days on SY food, the vials with singly housed females were checked for the presence of offspring. For each male, the number of females he fertilised during each of these seven days was recorded. We used 10 replicates per assay.

## Female egg-laying assay

Upon emerging as adults, male and female flies were transferred to fresh SY food and kept together for 48hours. Then female flies were then separated from males and kept on chemically defined diets with different AA ratios (FLYAA or FEMALEAA) and different total AA concentrations (0 g/L, 5.4 g/L or 10.7 g/L). Each vial contained five females, and flies were transferred

**Table 3. Amino Acid Stock Solutions (1000 mL).**

| FLYAA | | | MALEAA | | | FEMALEAA | | | | |
|---|---|---|---|---|---|---|---|---|---|---|
| AA | Amount | Units | AA | Amount | Units | AA | Amount | Units | Company | Code |
| ESSENTIALS | | | ESSENTIALS | | | ESSENTIALS | | | ESSENTIALS | |
| F | 5.04E-01 | g | F | 5.04E-01 | g | F | 4.69E-01 | g | Sigma-Aldrich | P2126-100G |
| H | 3.27E-01 | g | H | 2.80E-01 | g | H | 2.90E-01 | g | Sigma-Aldrich | H8000-100G |
| I | 5.61E-01 | g | I | 1.12E+00 | g | I | 1.08E+00 | g | Sigma-Aldrich | I2752-100G |
| K | 6.82E-01 | g | K | 8.86E-01 | g | K | 9.65E-01 | g | Sigma-Aldrich | L5626-100G |
| L | 1.02E+00 | g | L | 1.88E+00 | g | L | 1.78E+00 | g | Sigma-Aldrich | L8912-100G |
| M | 3.01E-01 | g | M | 2.96E-01 | g | M | 2.75E-01 | g | Sigma-Aldrich | M9625-100G |
| R | 8.14E-01 | g | R | 8.30E-01 | g | R | 9.04E-01 | g | Sigma-Aldrich | A5131-100G |
| T | 5.53E-01 | g | T | 4.99E-01 | g | T | 5.13E-01 | g | Sigma-Aldrich | T8625-100G |
| V | 5.99E-01 | g | V | 5.78E-01 | g | V | 5.83E-01 | g | Sigma-Aldrich | V0500-500G |
| W | 1.60E-01 | g | W | 1.82E-01 | g | W | 1.68E-01 | g | Sigma-Aldrich | T0254-100G |
| NON-ESSENTIALS | | | NON-ESSENTIALS | | | NON-ESSENTIALS | | | NON-ESSENTIALS | |
| A | 5.50E-01 | g | A | 5.49E-01 | g | A | 5.43E-01 | g | Sigma-Aldrich | A7627-100G |
| C | 1.71E-01 | g | C | 2.01E-01 | g | C | 2.43E-01 | g | Sigma-Aldrich | C7477-100G |
| D | 5.85E-01 | g | D | 6.58E-01 | g | D | 6.88E-01 | g | Sigma-Aldrich | A6683-100G |
| E | 7.59E-01 | g | E | 7.74E-01 | g | E | 8.12E-01 | g | Sigma-Aldrich | G5889-500G |
| G | 3.83E-01 | g | G | 3.96E-01 | g | G | 3.99E-01 | g | Sigma-Aldrich | G7126-100G |
| N | 5.14E-01 | g | N | 4.86E-01 | g | N | 4.78E-01 | g | Sigma-Aldrich | A0884-100G |
| P | 4.88E-01 | g | P | 4.66E-01 | g | P | 4.42E-01 | g | Sigma-Aldrich | P0380-100G |
| Q | 5.60E-01 | g | Q | 5.02E-01 | g | Q | 5.29E-01 | g | Sigma-Aldrich | G3126-100G |
| S | 6.88E-01 | g | S | 6.14E-01 | g | S | 6.07E-01 | g | Sigma-Aldrich | S4500-100G |
| Y | 4.64E-01 | g | Y | 8.40E-01 | g | Y | 8.45E-01 | g | Sigma-Aldrich | T8566-100G |

to fresh food every 24h for eight days. The numbers of eggs laid per vial per day for eight days were counted, using QuantiFly software (2.0), and recorded [63].

## Power analysis

Power analyses were performed to extract the confidence level (Z) of the fecundity assays used given the difference in fecundity between diets (E), the standard deviation observed (σ), and sample sizes used (n).

$$n = \left(\frac{Z\sigma}{E}\right)^2$$

## Calculating the exome and transcriptome matched aa proportions

The fly exome matched AA ratio (FLYAA) and the exome matched AA ratios of *Escherichia coli*, *Saccharomyces cerevisiae*, *Mus musculus*, *Zea mays*, and *Homo sapiens* were calculated as in [14]. Modifications to this procedure were performed as described below.

## Development of dietary transcriptome matched aa ratios

To estimate the transcriptome matched AA ratio, we computed the relative proportion (*P*) of each AA in the transcriptome, $P(AA_i)$ (where *i* indicates one of the 20 protein-coding AAs). This calculation was performed in two steps.

First, we calculated the number of instances of each AA encoded by each protein isoform, $AA_{ij}$ (where *j* indicates a protein isoform). This overcame the previous limitation in exome

**Table 4. Product amounts in 1000mL of Holidic Food.**

| Product | Amount | Unit | Company | Code |
|---|---|---|---|---|
| Agar | 2.00E+01 | g | Sigma-Aldrich | A7002-1KG |
| Sucrose | 1.71E+01 | g | Sigma-Aldrich | S1888-5KG |
| Acetic Acid | 3.00E+00 | mL | Thermo-Fisher | AJA1-2.5L GL |
| KH2PO4 | 3.00E+00 | g | Sigma-Aldrich | P9791-500G |
| NaHCO3 | 1.00E+00 | g | Sigma-Aldrich | S8875-500G |
| CaCl2 | 2.50E-01 | g | Sigma-Aldrich | C7902-500G |
| MgSO4 | 2.50E-01 | g | Sigma-Aldrich | M7506-500G |
| CuSO4 | 2.50E-03 | g | Sigma-Aldrich | C7631-250G |
| FeSO4 | 2.50E-02 | g | Sigma-Aldrich | F7002-250G |
| MnCl2 | 1.00E-03 | g | Sigma-Aldrich | M3634-100G |
| ZnSO4 | 2.50E-02 | g | Sigma-Aldrich | Z0251-100G |
| Choline Chloride | 5.00E-02 | g | Sigma-Aldrich | C1879-1KG |
| Myo-inositol | 5.00E-03 | g | Sigma-Aldrich | I7508-100G |
| Inosine | 6.50E-02 | g | Sigma-Aldrich | I4125-10G |
| Uridine | 6.00E-02 | g | Sigma-Aldrich | U3750-25G |
| Thiamine | 1.41E-03 | g | Sigma-Aldrich | T4625-5G |
| Riboflavin | 6.93E-04 | g | Sigma-Aldrich | R4500-5G |
| Nicotinic Acid | 8.38E-03 | g | Sigma-Aldrich | N4126-100G |
| Ca Pantothenate | 1.08E-02 | g | Sigma-Aldrich | 21210-5G-F |
| Pyridoxine | 1.74E-03 | g | Sigma-Aldrich | P9755-25G |
| Biotine | 1.41E-04 | g | Sigma-Aldrich | B4501-1G |
| Folic Acid | 5.00E-04 | g | Sigma-Aldrich | F7876-1G |
| Propionic Acid | 6.00E+00 | ml | Merck | 8.00605.0500 |
| Nipagin | 7.50E-01 | g | Sigma-Aldrich | W271004-5KG-K |
| Cholesterol | 3.00E-01 | g | Glentham | GE0100 |
| Absolute Ethanol* | 3.00E+01 | mL | Thermo-Fisher | AJA214-2.5LPL |
| Amino Acids | 1.07E+01 | g | See AA ratio table | |

*Solvent for cholesterol and nipagin

matching in which protein isoforms and length was not considered in the exome matched calculation.

$$AA_{ij} = |AA_{ij}|$$

To generate the transcriptome-weighted values for each AA ($AA_i$), we multiplied the number of each AA in each isoform, $AA_{ij}$, by the isoform's expression level, $E_j$ (measured in FPKM levels). We then summed the transcriptome-weighted AA abundance for each AA for all protein isoforms in the expressed genome.

$$AA_i = \sum_j (AA_{ij} * E_j)$$

To obtain the final proportion of each AA encoded by the transcriptome weighted exome $P(AA_i)$, we divided the total number of each type of AA, $AA_i$, by the sum of all transcriptome-weighted AAs, $\sum_i AA_i$.

$$P(AA_i) = \frac{AA_i}{\sum_i AA_i}$$

**Table 5. Molar Ratio Dietary Amino Acids.**

|  | Diet: | FLYAA | MALEAA | FEMALEAA |
|---|---|---|---|---|
| Essential Amino Acids | F | 0.037 | 0.039 | 0.036 |
|  | H | 0.026 | 0.023 | 0.024 |
|  | I | 0.052 | 0.056 | 0.054 |
|  | K | 0.057 | 0.062 | 0.068 |
|  | L | 0.094 | 0.093 | 0.088 |
|  | M | 0.025 | 0.026 | 0.024 |
|  | R | 0.057 | 0.051 | 0.055 |
|  | T | 0.056 | 0.054 | 0.055 |
|  | V | 0.062 | 0.063 | 0.064 |
|  | W | 0.010 | 0.012 | 0.011 |
| Non-essential Amino Acids | A | 0.075 | 0.079 | 0.078 |
|  | C | 0.017 | 0.019 | 0.016 |
|  | D | 0.053 | 0.049 | 0.051 |
|  | E | 0.063 | 0.059 | 0.062 |
|  | G | 0.062 | 0.068 | 0.068 |
|  | N | 0.047 | 0.047 | 0.047 |
|  | P | 0.052 | 0.052 | 0.049 |
|  | Q | 0.046 | 0.044 | 0.046 |
|  | S | 0.079 | 0.075 | 0.074 |
|  | Y | 0.031 | 0.030 | 0.030 |

Thus, the sum of the proportions of all transcriptome-weighted AAs was equal to one. We used 10 replicates per assay.

The dietary transcriptome matched AA ratios for male and female flies were obtained as the average AA ratio of five male and five female fly transcriptomes, three belonging to FlyAtlas 2 and two to ModEncode database (Table 1).

For all bioinformatics processing, we used R (3.4.4) and the R packages "seqinr" (3.4–5) and "stringr" (1.3.1) [64,65].

### Identification of limiting aas

The identification of the limiting AA, and the degree to which it was predicted limiting, was performed as described in [14].

### Tissue-specific data comparison

The AA ratio Cleveland plots generated to compare dietary AA ratios with tissue-specific transcriptome matched AA ratios were produced using the R package "ggplot2" (3.1.1) [66]. On these Cleveland plots, we identified the limiting AA in the exome (FLYAA) and the whole male (MALEAA) and female (FEMALEAA) transcriptome matched dietary AA ratios assuming that the tissue AA demand corresponded to the tissue-specific transcriptome matched AA ratios. The coverage value of each AA was calculated by dividing the molar proportion of the AA supplied in each diet (FEMALEAA, MALEAA, or FLYAA) by the molar proportion of each tissue-specific AA ratio. This ratio was expressed as a percentage value. For every tissue, the essential AA with the lowest percentage coverage was predicted to be limiting.

## Transcriptome data permutations

To calculate the transcriptome matched AA ratio, we sum the number of each AA in each gene multiplied by its expression calculated by FPKM. Then we divide the total count for each AA across all genes by the total number of all AAs across all genes. For each of the 20,000 permutations, we performed the same calculation after randomly assigning the expression weightings to genes in the genome.

## Genomes and transcriptomes

**Files and databases.**   Transcriptomic data for wild-type *Drosophila melanogaster* were obtained from Flyatlas 2, (European Nucleotide Archive, ENA, Study Accession: PRJEB22205) [31] and ModEncode (ENA Study Accession: SRP006203) [31,67]. The transcriptomes downloaded corresponded to RNAseq files from whole flies, 3 male samples and 3 female samples from Flyatlas 2, and 2 male samples and 2 female samples from ModEnconde, and tissue-specific RNAseq data from 3 male and 3 female fly samples (Flyatlas 2). Flyatlas 2 flies were of the strain Canton S and their transcriptomes were extracted 7 days post adult emergence [67]. ModEncode flies were of the strain [1] w[67c], and their transcriptomes were extracted 5–7 days post adult emergence [67]. The rearing conditions of the flies from FlyAtlas 2 and ModEncode are described in [31] and [67], respectively.

Transcriptomic data for *Drosophila melanogaster* (Canton S. strain) infected with *Escherichia coli* were obtained from [33] (Bioproject study accession: PRJNA428174). Each biological condition is the result of triplicate transcriptomes.

Reference genomes, gene annotations, and translated exomes were downloaded from the NCBI database (release: 210; *Escherichia coli*, *Saccharomyces cerevisiae*, *Mus musculus*, *Zea mays*, *Homo sapiens*) and Flybase (release: r6.21; *Drosophila melanogaster*).

**Transcriptome processing.**   RNAseq raw reads were trimmed for low-quality bases by Trimmomatic (0.38) [68] using default quality cutoff parameters (http://www.usadellab.org/cms/?page=trimmomatic). Any remaining rRNA reads were removed using SortmeRNA (4.2.0) [69].

Quality controls of the raw, trimmed, and aligned reads were performed by FastQC (0.11.17) and multiQC (v1.6) [70,71]. After trimming rRNA and low-quality reads, at least 4 million reads were obtained on each trimmed RNAseq sample with at least a 90% overall alignment rate and a 75% unique alignment rate with the reference genome. Quality control data is available on https://doi.org/10.26180/15048150.v1. Then, we used Hisat 2 (2.1.0) and Samtools (1.9), to map, index and sort the trimmed reads to the reference genome (Flybase release: r6.21) [72]. Transcript quantification was performed as Fragments per kilobase per million mapped read (FPKM) using Cufflinks (2.2.1) [73].

## Supporting information

**S1 Table. Detailed information of dietary components.**
(PDF)

**S2 Table. Composition of chemically defined diets.**
(PDF)

**S1 Fig. Proportional representation of amino acids from exome matching and transcriptome weighted exome matching profiles of Drosophila melanogaster challenged with an infection.**
(TIF)

**S2 Fig. Exome matching profiles from diverse species, including a bacterium, yeast, flies, corn, mouse, and human.**
(TIF)

## Acknowledgments

We thank Amy Dedman for her aid in research and laboratory management.

## Author Contributions

**Conceptualization:** Javier Gómez Ortega, David Raubenheimer, Christen K. Mirth, Matthew D. W. Piper.

**Formal analysis:** Javier Gómez Ortega, Christen K. Mirth, Matthew D. W. Piper.

**Investigation:** Javier Gómez Ortega.

**Methodology:** Javier Gómez Ortega.

**Supervision:** Sonika Tyagi, Christen K. Mirth, Matthew D. W. Piper.

**Writing – original draft:** Javier Gómez Ortega.

**Writing – review & editing:** Javier Gómez Ortega, David Raubenheimer, Sonika Tyagi, Christen K. Mirth, Matthew D. W. Piper.

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
