## [Decision Letter · Decision Letter 0]

15 Jun 2022

Dear Dr Piper,

Thank you very much for submitting your Research Article entitled 'The Biosynthetic Costs of Amino Acids at the Base of the Food Chain Determine Their Use in Higher-order Consumer Genomes' to PLOS Genetics.

The manuscript was fully evaluated at the editorial level and by independent peer reviewers. The reviewers appreciated the attention to an important problem, but raised some substantial concerns about the current manuscript. Based on the reviews, we will not be able to accept this version of the manuscript, but we would be willing to review a much-revised version. We cannot, of course, promise publication at that time.

If you decide to revise the manuscript for further consideration at PLOS Genetics, please aim to resubmit within the next 60 days, unless it will take extra time to address the concerns of the reviewers, in which case we would appreciate an expected resubmission date by email to plosgenetics@plos.org.

[LINK]

We are sorry that we cannot be more positive about your manuscript at this stage. Please do not hesitate to contact us if you have any concerns or questions.

Yours sincerely,

Gregory P. Copenhaver

Editor-in-Chief

PLOS Genetics

Reviewer's Responses to Questions

**Comments to the Authors:**

Reviewer #1: This manuscript represents an original study which tests whether the exome or the transcriptome is a better predictor of the amino acid requirements of adult Drosophila. The authors predicted that the transcriptome should be a better match to the amino acid requirements of specific sexes of flies than the average Drosophila ‘exome’. The manuscript calculates the amino acid requirements based on published transcriptomes from male and female flies. The authors test whether fecundity – the ultimate fitness measure - is influenced by the concentration of amino acids and the profile derived from the exome or the transcriptome. They found that the concentration of the amino acids available had a strong influence on the ability of males to fertilize females and of females to lay eggs. The transcriptome based diet, however, did not improve performance. They also identified the role of specific amino acids in the performance of males (tryptophan) and females (lysine). I found this experiment a fascinating extension of their previous work and a valuable contribution to the literature.

The authors also compared the AAs of 53 organisms including autotrophs (plants) and heterotrophs (animals, unicellular organisms). They concluded that animals had less diverse AA profiles than plants. They also concluded that ‘extremophiles’ were more constrained than ‘nonextremeophiles’. While I thought this was interesting, I also thought that Figures 5 and 6 were the weakest part of the manuscript.

I suggest that the analysis of the AAs of the exomes of the species shown in Figure 5 is conducted in a way that is a better fit with the rest of the manuscript. The data in this graph is quite biased towards heterotrophs, and it does not look like a linear model fits these data very well. I think that the authors should also provide more of a justification for the species chosen in this analysis. They need to better balance the representation of phyla and perhaps compare heterotrophs and autotrophs that are multicellular (or according to some other specific logic), rather than including unicellular organisms and comparing only heterotrophs and autotrophs. An analysis which did more to compare Drosophila to other insects and then to other animals would actually fit better in the context of the manuscript than this broad analysis of heterotrophs and autotrophs.

I also had difficulty believing the analysis of the data presented in Figure 6. Again, there is a strong data bias in the ‘non-extremophile’ group relative to the ‘extremophile’ group. This section also lacks a strong justification, and though the authors cite Krick et al. 2014 as the basis for the study of the metabolic cost of production of amino acids, I found it lacking enough detail for me to follow the arguments made in this paper.

Minor comments:

How many samples were used to calculate the transcriptome profiles used in these experiments? Were these transcriptomes of single flies or composites from pooled samples? Were several transcriptomes used, and at what time point in the animal’s life history were they collected? These details are missing from the manuscript but they are important to the interpretation of the results.

lines 390-393 - I don't follow the logic that heterotrophy 'constrains' evolution by limiting genome AA usage. Surely a counter example is aphids, which suck sap that is deficient in essential amino acids, and acquire needed amino acids from symbionts.

Reviewer #2: In this paper the authors report the effects of transcriptome-weighted exome matching on reproductive functioning in Drosophila. Relative to regular exome matching transcriptome weighting does not significantly improve reproductive function in either sex. Based on this result the authors argue that the efficiency of the use of dietary amino-acids is constrained at some pre-transcription level of biology. A cross-taxa synthesis reveals that the exome-matched composition of diets differs more among heterotrophs than autotrophs, and that the energetic costs of synthesising amino-acids explains their relative requirements.

This is an interesting paper that has been well prepared for publication. The paper reads well and guides the reader through the logic of the different steps in the analysis. It makes clever use of data. I have some comments that the authors may wish to take into consideration. Generally, I am supportive of publication.

L120-122. Are the authors at all concerned about the imprecision/specificities of the expression data they are using? Presumably the gene expression data used are an estimate from an experiment, which come with standard errors etc and will have been measured in different animals. I know the authors can only use what is available and I am not suggesting they should have done anything else, but I think this is a point that should perhaps be addressed in the discussion.

L189. “another diet in which tryptophan was omitted” – was this the rich FLYAA diet with tryptophan removed? If so, I would specify that, if not, what was it?

L211. RE salivary glands, could this be an alternative explanation for your results? If both MALEAA and FEMALEAA are deficient for the salivary glands is food intake impacted on these diets, meaning that in fact intake of amino-acids is impacted relative to FLYAA?

L352-354. This is a neat way of seeing how the exome-matched diets compare in context. I wondered whether adding the composition of yeast also helps add this context? For reference, for example, are all three exome matched diets in Fig 1 more similar to one another than to some commonly used yeast (e.g., in the SY diet)?

L376. I found the wording of this sentence a bit strange. Seems more natural to say “… the transcriptome affects/dictates the consumer’s AA requirements.”

Figure 5. This looks like regression through the origin here. You need to be very careful that these models pass a validation step. Often regression through the origin makes sense biologically, but violates model assumptions. In the case of analysis its often better to sacrifice the former in favour of the latter.

L405-413. Two points here. 1) Revealing my ignorance about how the cost of calculation is made, here, but isn’t the differential cost to extremeophiles (e.g., due to temperature) factored in, or is a common method used? 2) It would be nice if it were possible to include a directional prediction? Do expect all extremophiles to differ in the same direction?

Figure 6. This regression/scatter plot seems upside down to me.

L503-512. I really struggled with this, even after having read it several times. I suspect may other readers would as well. Can the authors bring more clarity to what is being conveyed.

Reviewer #3: This manuscript by Ortega et al concerns the match between dietary amino acid supply and user need, using a Drosophila model. This is the latest in a string of interesting and important papers from the authors in this area.

The paper comes in two halves. The first half builds on published results showing that in silico translation of the exome predicts a ratio of dietary amino acids that better matches consumer need. In the current manuscript, the authors ask if weighting the exome match algorithm with transcriptome data can improve the prediction of optimal diet. This extension of the published protocol is not successful in the current study, with results showing that diets matched to sexual dimorphism in the transcriptome do not enhance the fitness of either sex. The second half of the paper goes on to show that in silico translation of exomes of organisms across higher trophic levels yields patterns of amino acid use that are predicted by costs of biosynthesis in primary producers. Throughout, data appear to have been collected at the highest standard, analysis is appropriate, writing is clear, and dataviz is outstanding.

The second half of the paper is extremely interesting, novel, and has potentially major implications for our understanding of how nutritional physiology and ecology evolve. I believe this second half alone would attract significant interest if published in PLoS Genetics. Indeed the paper's title refers only to this half. I have only one question about this part of the paper, for discussion: if AA usage regresses to cost of production, there should be close relationship between AA ratios across organisms. Why then in previous work was yeastAA shown to be sub-optimal relative to flyAA? Is the difference between these organisms basically residual variation away from the mean(s)?

My enthusiasm for the first half of the paper is somewhat more limited: it is not clear what these new results add to the field, and the paper's title does not mention them. Basically, the implicit conclusion is that exome matching doesn't work, but previous results show unambiguously that it can. Transcriptome-weighted exome matching might work in other contexts, e.g. stress resistance. This means that the second part of the paper, which does contain new findings, is not well set up by the first, and indeed the jump from matching diets to transcriptomes is a bit sudden, without progressive logic. It is clearly important to be able to publish negative results, and it's clear that a lot of work went into these experiments, but in this case without some forms of positive controls, some alternative explanations for the (unfortunate) lack of signal are possible. My questions are:

Could males and females just be so different that fitness assays are not comparable? Might sex-matched diets improve other traits that could be measured in equivalently in each sex but weren't measured here? e.g. xenobiotic or starvation resistance?

Are the FlyAtlas data sufficiently granular to reveal the reproduction-limiting cells in each sex - Do we know which reproductive tissue is protein-limited? FlyAtlas characterized many tissues, but there is still a lot of uncharacterized complexity within a tissue, and it can take only a very few cells to limit overall function.

Might the sexes even differ in which tissues are limiting? e.g. could male fitness not be a function of courtship and therefore function of particular neurons, which might not limit females in the same way?

The male fitness assay measures number of inseminated females, which is influenced by behavior. Might number of offspring per inseminated female have been a more direct measure of sperm quality, indicative of nutritional quality?

The FlyAtlas data were collected in different flies, on a different diet. Both the male and female transcriptome were likely different from what they would be on FLYAA, which is the real baseline we would need for transcriptome weighting to improve upon. Might confounding variation have clouded impact of transcriptome weighting?

In summary, I enjoyed reading the paper, I believe it will be of interest to the readership of PLoS Genetics, and will make an important impact on multiple fields. However I think the important parts aggregate in the second half.

**Have all data underlying the figures and results presented in the manuscript been provided?**

Reviewer #1: Yes

Reviewer #2: **No: **I'm not sure the cross-taxa data are available.

Reviewer #3: Yes

PLOS authors have the option to publish the peer review history of their article (what does this mean?). If published, this will include your full peer review and any attached files.

Reviewer #1: No

Reviewer #2: No

Reviewer #3: No

---

## [Decision Letter · Decision Letter 1]

14 Dec 2022

Dear Dr Piper,

Thank you very much for submitting your Research Article entitled 'BIOSYNTHETIC CONSTRAINTS ON AMINO ACID SYNTHESIS AT THE BASE OF THE FOOD CHAIN MAY DETERMINE THEIR USE IN HIGHER-ORDER CONSUMER GENOMES' to PLOS Genetics.

The manuscript was fully evaluated at the editorial level and by two of the original independent peer reviewers. One reviewer is now satisfied (but notes a couple of minor typos).  The other reviewer still has fundamental concerns about the first half of the manuscript.  If possible, addressing their concerns experimentally would be optimal, but if that is not feasible, I believe it would be sufficient to explicitly state the limitations/caveats of the conclusions drawn from that data in the text of the manuscript (as described by the reviewer) - that would allow readers to weight those conclusions appropriately.  

Based on the reviews, we will not be able to accept this version of the manuscript, but we would be willing to review a much-revised version. We cannot, of course, promise publication at that time.

If you decide to revise the manuscript for further consideration at PLOS Genetics, please aim to resubmit within the next 60 days, unless it will take extra time to address the concerns of the reviewers, in which case we would appreciate an expected resubmission date by email to plosgenetics@plos.org.

We are sorry that we cannot be more positive about your manuscript at this stage. Please do not hesitate to contact us if you have any concerns or questions.

Yours sincerely,

Gregory P. Copenhaver

Editor-in-Chief

PLOS Genetics

Reviewer's Responses to Questions

**Comments to the Authors:**

Reviewer #2: The authors have addressed all of my comments. Two minor typos I saw:

L42: Double “that”

L50: “protein limitation”: i.e., not plural

Reviewer #3: I have re-read the manuscript and unfortunately I do not feel that the revisions adequately address the last round of comments.

The second half of the paper (biosynthetic constraints of AA production) is publishable alone in PLoS genetics, independent of the first half of the paper. However the first half fundamentally delivers a null result, but since the assays of males and females are different, we don't know if lack of response to transcriptome weighting is a true negative - we might just be measuring different things. For this reason an assay that can be conducted equivalently before and after transcriptome weighting would be more appropriate. Assays of immunity before and after weighting would indeed be very interesting, and I recognise the authors' efforts to include some data on this. However without functional immune assays to show whether flies fed weighted/unweighted diets are more resistant to infection, we cannot make any conclusions. Showing that immune response-weighted diets are predicted to be less different than sex-weighted diets does not mean that the assays of males and females reproduction are definitely measuring the same thing.

My recommendation is still that the second half of the paper is a solid publishable unit, but the first half does not convince me that transcriptome-weighted diets should not be pursued.

**Have all data underlying the figures and results presented in the manuscript been provided?**

Reviewer #2: Yes

Reviewer #3: Yes

PLOS authors have the option to publish the peer review history of their article (what does this mean?). If published, this will include your full peer review and any attached files.

Reviewer #2: No

Reviewer #3: No

---

## [Decision Letter · Decision Letter 2]

24 Jan 2023

Dear Dr Piper,

We are pleased to inform you that your manuscript entitled "BIOSYNTHETIC CONSTRAINTS ON AMINO ACID SYNTHESIS AT THE BASE OF THE FOOD CHAIN MAY DETERMINE THEIR USE IN HIGHER-ORDER CONSUMER GENOMES" has been editorially accepted for publication in PLOS Genetics. Congratulations!

Yours sincerely,

Gregory P. Copenhaver

Editor-in-Chief

PLOS Genetics

Comments from the reviewers (if applicable):

Reviewer's Responses to Questions

**Comments to the Authors:**

Reviewer #2: The authors have addressed all of my comments

Reviewer #3: Acknowledging the limitations of the assays is a fair strategy. This is an important piece of work and I am happy to recommend publication.

**Have all data underlying the figures and results presented in the manuscript been provided?**

Reviewer #2: None

Reviewer #3: Yes

PLOS authors have the option to publish the peer review history of their article (what does this mean?). If published, this will include your full peer review and any attached files.

Reviewer #2: No

Reviewer #3: No

**Data Deposition**

http://datadryad.org/submit?journalID=pgenetics&manu=PGENETICS-D-22-00575R2

**Press Queries**

---

## [Editor Report · Acceptance letter]

9 Feb 2023

PGENETICS-D-22-00575R2 

BIOSYNTHETIC CONSTRAINTS ON AMINO ACID SYNTHESIS AT THE BASE OF THE FOOD CHAIN MAY DETERMINE THEIR USE IN HIGHER-ORDER CONSUMER GENOMES 

Dear Dr Piper, 

We are pleased to inform you that your manuscript entitled "BIOSYNTHETIC CONSTRAINTS ON AMINO ACID SYNTHESIS AT THE BASE OF THE FOOD CHAIN MAY DETERMINE THEIR USE IN HIGHER-ORDER CONSUMER GENOMES" has been formally accepted for publication in PLOS Genetics! Your manuscript is now with our production department and you will be notified of the publication date in due course.

With kind regards,

Zsuzsanna Gémesi

PLOS Genetics

On behalf of:
